# SELFI: Autonomous Self-Improvement with RL for Vision-Based Navigation around People

**Noriaki Hirose**[1,2], **Dhruv Shah**[1], **Kyle Stachowicz**[1], **Ajay Sridhar**[1], **Sergey Levine**[1]
[1] University of California Berkeley    [2] Toyota Motor North America
noriaki.hirose@berkeley.edu

**Abstract:** Autonomous self-improving robots that interact and improve with experience are key to the real-world deployment of robotic systems. In this paper, we propose an online learning method, SELFI, that leverages *online* robot experience to rapidly fine-tune pre-trained control policies efficiently. SELFI applies online model-free reinforcement learning on top of offline model-based learning to bring out the best parts of both learning paradigms. Specifically, SELFI stabilizes the online learning process by incorporating the same model-based learning objective from offline pre-training into the Q-values learned with online model-free reinforcement learning. We evaluate SELFI in multiple real-world environments and report improvements in terms of collision avoidance, as well as more socially compliant behavior, measured by a human user study. SELFI enables us to quickly learn useful robotic behaviors with less human interventions such as pre-emptive behavior for the pedestrians, collision avoidance for small and transparent objects, and avoiding travel on uneven floor surfaces. We provide supplementary videos to demonstrate the performance of our fine-tuned policy on our project page [1].

**Keywords:** online reinforcement learning, vision-based navigation

## 1 Introduction

Reinforcement learning (RL) provides an appealing algorithmic approach for autonomously improving robotic policies in unpredictable and complex real-world settings [1–4]. For example, in the indoor navigation scenario depicted in Fig. 1, the robot needs to not only avoid obstacles, but also deal with unpredictable and hard-to-model situations, like the interaction with the pedestrian. Model-based control methods can struggle with the unpredictable elements in the scene [5–7], and RL in principle provides an appealing alternative: learn directly from real-world experience, sidestepping the need for highly accurate modeling. However, directly performing end-to-end RL from scratch in the real world can be difficult: discovering a high-quality policy may require a large number of trials and encounter catastrophic failures during the training process [8–10]. This is especially problematic when it is not possible to provide external instrumentation that avoids catastrophic failures — for example, with human interactions where failures might be inconvenient or even dangerous.

In this work, we propose a framework for robotic learning that aims to address this challenge by utilizing model-free RL fine-tuning on top of a learning-enabled model-based policy that is pre-computed offline. Our approach initializes the robot from a policy that already exhibits basic competency in its environment, and from there further improves its behavior *in the particular setting where it is situated* through trial-and-error. While a number of prior works have examined the use of model-free RL as a fine-tuning strategy on top of pre-trained or pre-computed policies [11–14], this is often complicated by the fact that initializing sample-efficient model-free RL methods requires not only a policy but also a critic [3, 4, 15–18]. The policy initialization can often be derived from a prior policy (either classic or learned), but in modern actor-critic methods, the policy is trained

---

[1] sites.google.com/view/selfi-rl

8th Conference on Robot Learning (CoRL 2024), Munich, Germany.

rapidly to maximize the critic's value, and this quickly overrides any actor initialization if the critic is not also pre-trained [3, 4, 17, 18]. Our key observation is that model-based methods that maximize some sort of planning objective can *also* be used to initialize the critic, such that model-free RL Q-values are given by a linear combination of a learned model-free critic and a model-based trajectory value estimate. In this design, as long as the critic is initialized to produce small values, the RL process starts off by maximizing the model-based value estimate (i.e., model-based control), and then improves further through trial-and-error interaction.

We illustrate this design in Fig. 1, and we instantiate our system in the context of social navigation: the problem of navigating an indoor space while avoiding undesirable behavior around pedestrians, such as interruptions, collisions, or invasion of their intimate distance. This problem domain is a good fit for validating our framework because model-based policies can be derived from geometric models of the world and rough predictive models of pedestrians, but

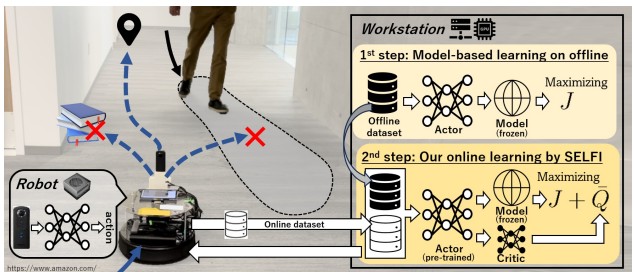

Figure 1: **Overview of our proposed online learning system, SELFI**. Our method fine-tunes a pre-trained control policy trained with model-based objective by incorporating this objective into a Q-value function to maximize during online model-free RL.

these policies can be significantly improved through model-free trial-and-error, both because the pedestrian models might be inaccurate, and because the robot can adapt directly to the behavior of the pedestrians in a specific downstream deployment environment. In this setting, our model-based policies are derived from the previously proposed SACSoN [5], which constructs policies by optimizing a trajectory value using a 3D reconstruction of training environments and predictive models of humans. This model-based procedure also provides trajectory value estimates that can bootstrap the model-free RL critic. During the real-world model-free RL phase, we improve the model-based policy by learning a residual value critic and applying actor-critic methods as described above.

The main contributions of this paper is to propose a framework, SELFI, that takes advantage of the best aspects of online RL and offline model-based learning. Specifically, SELFI uses online Q-learning to fine-tune a control policy trained with offline model-based learning. SELFI rapidly improves the performance of a pre-trained policy in the target environment without needing significant human intervention during online learning. In our evaluations, SELFI improves the performace of the pre-trained policy in multiple vision-based navigation tasks, greatly outperforming policies trained purely offline as well as standard end-to-end model-free offline-to-online RL fine-tuning methods. Within only two hours of fine-tuning, our policy learns complex robotic behaviors, e.g., pre-emptive behavior for navigating around pedestrians, collision avoidance for unseen small and transparent obstacles, and preferences for smooth and easily traversable surfaces. The robotic behaviors learned by SELFI are shown in the supplemental videos.

## 2 Related Work

We review the related learning methods as well as the navigation methods on the evaluation task.

**Online learning:** There are various data-driven methods for adapting control policies to their real-world environments through interactions. In learning-based settings, DAgger [19] is a general framework that iteratively trains a control policy with expert labeled demonstrations. Additionally, model-based learning can fine-tune control policies by utilizing differentiable dynamic forward models to define an objective function with visual foresight [20–22], reward prediction [23], and state prediction [5, 24, 25]. This type of learning can be combined with an optimization algorithm to generate action commands online, just as in model predictive control [26–29]. However, for any model-based approach, the performance of the learned control policy is limited to the accuracy of the model and the quality of the dataset. Model-based RL, which learns a model from interactions with the real world, is subject to similar limitations [30–32].

Model-free RL accumulates data, including rewards from real-world interactions, and trains a control policy to maximize the expected sum of discounted rewards from future timesteps [1–4]. Although model-free RL does not suffer from modeling errors, running model-free RL from scratch requires a significant amount of time for for data collection and learning. In addition, certain interactions with the environment can be dangerous for the robot itself and humans. Prior work has studied how offline learning addresses this issue by pre-training policies in simulation [33, 34] or in the real world using behavior cloning [25, 35, 36] or offline RL [11, 12, 37, 38]. Hybrid approaches use a learned dynamics model or a control policy learned with model-based RL to initialize a model-free learner, making it more sample efficient [39, 40].

Our proposed approach is closely related to Residual RL [14], which decomposes the policy's output into a solution from an existing controller and the actions from a residual policy trained with model-free RL. By leveraging the existing controller, residual RL stabilizes the robot's behavior during the early stages of online learning and learns the target behavior via the flexibility of model-free RL. However, the confusion between the existing and learned control policies due to composing them in *the action space* restricts the performance of residual RL. Different from these previous works, our proposed method, SELFI, seamlessly composes model-based learning and model-free RL in *the objective space*. SELFI is a flexible and stable method for online fine-tuning with model-free RL because it incorporates the objective used in offline model-based learning in the learned Q-values.

**Social navigation:** Social navigation has been extensively explored in [41–43]. Model-based approaches relying on dynamic pedestrian models have been utilized for behavior modeling [6, 7, 44–46]. Many existing techniques determine the robot's actions based on predicted pedestrian behavior [47–53]. Furthermore, social navigation has also been explored using model-free learning approaches, such as reinforcement learning [54–60]. Unlike vision-based navigation [21, 25, 61–63] as our evaluation task, which only uses RGB camera observations, these approaches rely on detected pedestrian poses and/or multiple metric sensors such as LiDARs and/or depth cameras [64, 65].

The most related work, SACSoN [5], utilizes the model-based learning architecture to learn a socially unobtrusive policy in vision-based navigation setting. However, even simple socially compliant behaviors are still challenging to learn due to the modeling errors. Hence, we apply our proposed method, SELFI, to fine-tune the pre-trained SACSoN policy to improve its performance. We conduct comparisons with baseline methods as well as other online fine-tuning methods in the same task setting, vision-based navigation as our method.

## 3 Combining Model-based Control with Online Model-Free RL

### 3.1 Preliminaries

We apply our hybrid model-based and model-free learning algorithm to a Markov Decision Process (MDP) $\mathcal{M}$. We first briefly explain model-free RL and model-based learning, respectively.

**Model-free RL:** In RL, we want to maximize the expected sum of discounted rewards. $Q$-learning algorithms [4, 66] solve this task by learning a function approximating $Q^{\pi_\theta}(s,a) = \mathbb{E}_{\{s_t,a_t\}\sim\mathcal{M}^\pi} \sum_{t=0}^\infty \gamma^t r(s_t, a_t)$ where trajectories with states $s_t$ and actions $a_t$ are sampled from the closed-loop dynamics of $\mathcal{M}$ under the policy $\pi_\theta$. $\gamma$ indicates the discount factor for future rewards $r$. Here, mathematical symbols without a subscript representing time show the current time state and action, e.g., $s = s_0$, $a = a_0$. The $Q$-function for the optimal policy obeys the Bellman equation, and it can be trained to minimize the TD error $\delta = r(s,a) + \gamma \max_{a'} Q^{\pi_\theta}(s', a') - Q^{\pi_\theta}(s, a)$ where $s'$ and $a'$ indicate the next step $s$ and $a$, respectively. In the actor-critic setting, we learn both an approximation for the action-value function $Q^{\pi_\theta}(s, a)$ and for the policy that maximizes the action-value function $Q(s, a)$ as $\pi_\theta(s)$. This enables tractable optimization over large action spaces where the $\max$ in the Bellman equation cannot be efficiently computed.

**Model-based learning:** In the model-based learning setting, we optimize an objective over open-loop *sequences* of virtual actions $\tau = \{\hat{a}_t\}_{t=0...H-1}$ using an approximate dynamics model and

reward estimate as $\arg\max_{\tau=\{\hat{a}_t\}} \left[ J(s,\tau) := \sum_{t=0}^{H-1} \gamma^t \hat{r}(\hat{s}_t, \hat{a}_t) \right]$ s.t. $\hat{s}_{t+1} = \hat{f}(\hat{s}_t, \hat{a}_t)$ where $\hat{s}_t$ is the predicted state under the approximate dynamics described by $\hat{f}$, which can be either learned or given. $\hat{a}_t$ is the $t-$th virtual action, which is estimated at the current time state. Note that $\hat{s}_0 = s$ and $\hat{a}_0 = a$ in $J(s,\tau)$. We assume that the dynamics $\hat{s}_{t+1} = \hat{f}(\hat{s}_t, \hat{a}_t)$ and rewards $\hat{r}(\hat{s}_t, \hat{a}_t)$ are easy to compute and differentiable, allowing us to directly optimize the sequence of actions online with gradient-based trajectory optimization [26, 28]. Instead of optimizing this objective online (analogous to traditional nonlinear model-predictive control methods), model-based learning amortizes this optimization by learning the parameters $\theta$ of a control policy $\tau = \pi_\theta(s)$ to optimize this objective offline. This distills the optimization problem into an *offline base policy* $\pi_\theta$, represented as a neural network mapping observations to sequences of actions $\tau$. At runtime, actions are then sampled from the learned policy $\pi_\theta$. A number of prior methods have proposed similar methods for model-based learning [20, 21, 23, 25].

## 3.2 SELFI learning architecture

In SELFI, we combine the strengths of model-based learning with the strengths of model-free learning to enable finetuning in the real world. We wish to decompose the critic value $Q$ into a model-based objective $J$ and a learned residual objective $\bar{Q}$ as $Q(s,\tau) = J(s,\tau) + \bar{Q}(s,a)$ where $a$ is the first action $\hat{a}_0$ in the sequence of virtual actions $\tau = \{\hat{a}_0, \ldots, \hat{a}_{H-1}\}$. We train $\bar{Q}(s,a)$, which corresponds to the part of the overall objective that is not considered by model-based learning. We fine-tune the pre-trained control policy $\pi_\theta(s)$ to maximize the combined Q-function $Q(s,\tau)$. Accordingly, we first assume $r(s_t, a_t)$ in timesteps $t = 0 \ldots H-1$ has the form, $r(s_t, a_t) = \hat{r}(\hat{s}_t, \hat{a}_t) + \bar{r}(s_t, a_t)$ where $\hat{r}(\hat{s}_t, \hat{a}_t)$ is the model-based reward and $\bar{r}(s_t, a_t)$ is the unmodeled residual reward. Hence, we have:

$$Q(s,\tau) = \sum_{t=0}^{\infty} \gamma^t r(s_t, a_t) = \underbrace{\sum_{t=0}^{H-1} \gamma^t \hat{r}(s_t, a_t)}_{J(s,\tau)} + \underbrace{\sum_{t=0}^{H-1} \gamma^t \bar{r}(s_t, a_t) + \sum_{t=H}^{\infty} \gamma^t r(s_t, a_t)}_{\bar{Q}(s,a)}. \quad (1)$$

The model-free $\bar{Q}$ includes i) the reward terms that cannot be directly modeled and ii) the value of the long-horizon returns that are ignored by the limited-horizon model-based learning. As usual, $Q(s,\tau)$ should obey the Bellman equation $Q(s,\tau) = r(s,a) + \gamma Q(s', \pi_\theta(s'))$ and the critic for $\bar{Q}(s,a)$ can be trained to minimize the TD error $\tilde{\delta}$:

$$y \leftarrow r(s,a) + \gamma Q(s', \pi_\theta(s')); \quad \tilde{\delta} = y - Q(s,\tau). \quad (2)$$

In practice we use a delayed copy of $\bar{Q}$ as a *target network* to compute target values [18]. During online learning, we calculate the gradient of $\pi_\theta$ by back-propagation to maximize the hybrid objective $Q(s,\tau)$. By including the model-based objective term $J(s,\tau)$ in $Q(s,\tau)$, the initial estimate of $Q(s,\tau)$ already provides reasonable values with a suitably initialized $\bar{Q}(s,a)$ (e.g., with small initial weights), which significantly stabilizes performance early in training. Then, $\bar{Q}(s,a)$ can be further trained with online interactions to learn the target robotic behaviors.

## 3.3 SELFI implementation

As our underlying RL algorithm, we use a variant of twin delayed deep deterministic policy gradients (TD3) [18], which is a sample-efficient and stable algorithm for training deterministic control policies. Fig. 1 shows the system overview of SELFI for real-world learning. Similar to [13], policy training is implemented on an workstation while the robot's onboard computer is used for inference.

On the workstation, we train with batch data from both the offline and online dataset to avoid overfitting to the online data. Half of the batch data is chosen from the offline dataset, and the other half is from online data, which is collected by the robot. We update the actor once for every two updates to the critic, maximizing $Q(s,\tau) = J(s,\tau) + \bar{Q}(s,a)$ with respect to the policy.

On the robot, we calculate the learned policy to obtain the sequence of actions $\tau = \{\hat{a}_t\}_{t=0\ldots H-1}$ and execute the first action $\hat{a}_0$, adding Gaussian noise $\epsilon$ to encourage exploration. Then, the robot

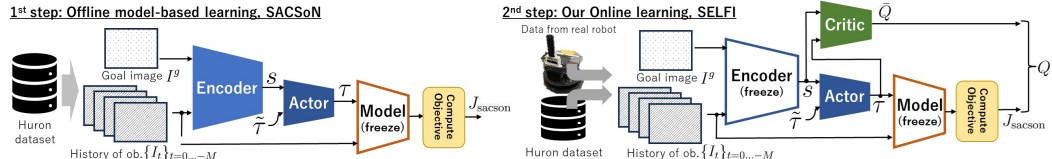

Figure 2: **SELFI architecture overview.** Before online learning, we train the encoder and the actor by maximizing the differentiable model-based objective. In the online phase, we combine the offline objective with the learned $Q$-value from model-free RL to fine-tune the actor.

sends the collected data to the workstation, where it is stored in the replay buffer. Periodically, the updated actor weights are sent from the workstation to the robot so it can use the latest policy.

## 4    SELFI System Setup

We evaluate SELFI on for a vision-based social navigation task, where a robot navigates an indoor environment with pedestrians. We employ SACSoN [5] as the offline model-based objective. In the online phase, SELFI fine-tunes the pre-trained policy to learn socially-compliant behavior including 1) pre-emptive avoidance of oncoming pedestrians, 2) collision avoidance for the small or transparent objects, and 3) avoiding travel on uneven floor surfaces. These behaviors are difficult to learn purely from offline model-based learning due to the modeling errors and insufficient information in the offline dataset. In this section, we describe the implementation of SELFI on top of SACSoN.

### 4.1    Offline Learning with SACSoN

We briefly describe the learning procedure in SACSoN. Details are shown in the original paper [5].

**Offline model-based objectives:** As shown in Fig. 2, we represent the control policy by an encoder $g_\phi$ coupled to an actor network $\pi_\theta$. We train the policy to maximize the model-based objective [5], $J_{\mathrm{sacson}}(s, \tau) := \sum_{t=0}^{H-1} \hat{r}_t^{\mathrm{pose}} + \hat{r}_t^{\mathrm{geom}} + \hat{r}_t^{\mathrm{ped}} + \hat{r}_t^{\mathrm{reg}}$ where the $\hat{r}^{\mathrm{pose}}$ reward encourages goal-reaching behavior, $\hat{r}^{\mathrm{geom}}$ [25] penalizes collision with static obstacles (via signed point-cloud distance), $\hat{r}^{\mathrm{ped}}$ [5] is to learn socially unobtrusive behavior and $\hat{r}^{\mathrm{reg}}$ acts as a regularization term to encourage smooth motion. All objectives in $J_{\mathrm{sacson}}$ are differentiable with respect to the action sequence $\tau$ and can therefore calculate the gradient of $g_\phi$ and $\pi_\theta$ via $\tau$ and learn them.

**Offline training:** The offline policy $\pi_\theta$ and $g_\phi$ are trained on the 80-hour HuRoN dataset [5] for vision-based navigation including over 4000 human-robot interactions. Observations consist of a 2-second sequence of six 128×256 omnidirectional camera images from a Ricoh Theta S, together with the goal image $I^g$, and predict a sequence of eight future actions. To allow the critic to handle any actuator delays in the system, we concatenate the previous action with the extracted image features as shown in Fig. 2.

### 4.2    Online Learning with SELFI

We demonstrate an instantiation of SELFI for the socially-compliant navigation, following Sec. 3.2 and Sec. 3.3. Here, we describe the specific procedure we use for online RL finetuning.

**Learning setting:** To obtain the target robotic behavior as fast as possible during online learning, SELFI fine-tunes the actor $\pi_\theta$ with a frozen encoder $g_\phi$ as shown in Fig. 2. It corresponds to defining the extracted feature from the encoder as $s$ in formulation of SELFI. To estimate $\bar{Q}$, we feed the feature $s$ and the sequence of action $\tau$ to the critic, as shown in right part of Fig. 2. By sharing the encoder with the actor, we efficiently learn both actor and critic. In the TD error calculation, we assume $J(s, \tau) \approx \gamma J(s', \tau')$ to train the small critic network during the brief online learning phase.

**Reward design:** The reward is designed as $r(s, a) = \vec{v} \cdot \hat{g} + C_s + C_d$ to incentivize smoothness and progress towards the goal, while avoiding collisions: The first term maximizes velocity towards the next goal, where $\hat{g} := [x_g, y_g, \theta_g]$ and $\vec{v} := [v, 0, \omega]$ are goal direction and velocity vectors

Table 1: **Closed-loop Evaluation of trained control policies.** IDV is intimate distance violation duration, NCO is near-collision duration, and UFS is duration on uneven floor surface, CP is the number of collision for pedestrians, CO is the number of collision for the tiny objects, Int is the number of interventions by teleoperators, SPL is Success weighted by Path Length [70] and STL is Success weighted by Time Length [71]. SPL and STL are calculated by assuming that the robot reaches the goal position even regardless of there being a human intervention. $*$ indicates using the ground truth goal pose for generating the velocity commands.

| Method | IDV [s] $\downarrow$ | NCO [s] $\downarrow$ | UFS [s] $\downarrow$ | CP $\downarrow$ [#] | CO $\downarrow$ [#] | Int $\downarrow$ [#] | SPL $\uparrow$ | STL $\uparrow$ |
|---|---|---|---|---|---|---|---|---|
| Sampling-based motion planning$*$ | 21.512 | 11.633 | 10.212 | 1.333 | 3.333 | 0.733 | 0.808 | 0.652 |
| Residual RL | 28.731 | 12.365 | 8.073 | 3.200 | 4.333 | 2.067 | 0.817 | 0.641 |
| SACSoN with fine-tuning | 18.018 | 11.229 | 3.578 | 0.800 | 2.667 | 0.733 | 0.838 | 0.596 |
| Ours | **7.978** | **3.938** | **3.067** | **0.200** | **1.400** | **0.267** | **0.918** | **0.739** |

expressed in the robot's current frame [13]. To obtain the local goal pose $g$, we build an approximate localization system that incorporates visual odometry and AR markers along the robot's trajectory. The specific choice of AR markers is a design decision to facilitate easy online learning, and other mechanisms for localization based on visual odometry(see Appendix F). $v$ and $\omega$ are the linear and the angular velocity commands for the two-wheel-drive robot in $\hat{a}_0$. We use $C_s$ and $C_d$ to denote the rewards for avoiding static obstacles and dynamic obstacles (pedestrians) respectively:

$$C_s = \begin{cases} -C_c & (\text{if } collision \text{ is True}) \\ -C_b & (\text{else if } bumpy \text{ is True}) \\ 0.0 & (\text{otherwise}) \end{cases} \qquad C_d = \begin{cases} -C_h & (\text{if } d_h < 0.5 + r_r) \\ 0.0 & (\text{otherwise}) \end{cases} . \quad (3)$$

Here, $C_c$, $C_b$, and $C_h$ are positive constants that penalize undesirable behaviors. We set $C_c = C_b = 0.3$ and $C_h = 0.1$, and do not tune these values for our experiments.

The robot triggers a *collision* event using the robot's bumper sensor, and a *bumpy* event (caused by an uneven floor) when the measured acceleration exceeds a fixed threshold. Note that these sensors are not mandatory and can be substituted by the other sensors commonly used in navigation [13]. To detect intimate distance violations, we estimate the distance $d_h$ to the closest pedestrians using a combination of semantic segmentation [67, 68] and monocular depth estimation [69]. $r_r = 0.5$ m is the radius of the circular robot footprint with margin.

**Others:** We set the discount factor $\gamma = 0.97$, accounting for long trajectories of human-robot interaction behaviors. In addition, the workstation sends the policy model parameters $\theta$ to the robot every 50 training steps (approx. 1 minute wall clock time). Since we conduct online learning for maximum of two hours, the maximum number of training steps is about 6000. On online learning, we set the batch size as 76. The learning rate of Adam optimizer is set as 0.0001. All other parameters follow the authors' implementation of TD3 [18] and SACSoN [5].

## 5 Evaluation

Our experiments evaluate SELFI in the real world, studying the following research questions:

  **Q1.** Does SELFI lead to better final policy performance than existing approaches?
  **Q2.** Does SELFI reduce interventions during the fine-tuning process (without degradation)?

Please see Appendix A for more details on a prototype robot and whole navigation system for our evaluation. And we show the details of our online training and evaluation setup in Appendix B.

**Performance analysis of the fine-tuned control policy:** We compare to the strongest baselines, **Residual RL** and **SACSoN with fine-tuning**. We fine-tune our method as well as these baselines and evaluate the fine-tuned policy. In addition, we evaluate **Sampling-based motion planning** as the classical motion planning. The details of these baseline methods and the online training setups are shown in Appendix B and G. In evaluation, it is difficult to conduct extensive and reproducible comparisons in the highly populated natural environments, because they are uncontrolled, with pedestrians walking in and out of the scene at random. Therefore, to conduct a more controlled and reproducible comparison with each of the baselines, addressing **Q1**, we focus on the "organized" environments specifically.

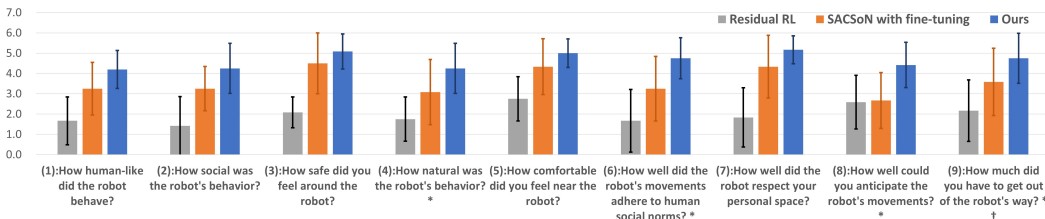

Figure 3: **Evaluation of socialness by human rating.** The height of each bar indicates the mean and the range line indicates the standard deviation. Larger is better in all ratings. ∗ indicates the statistical significance of our method on the t-test with $p < 0.05$. † indicates scale reversed for analyses.

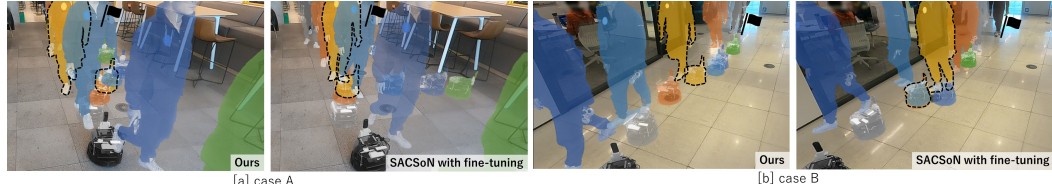

Figure 4: **Visualization of the robot behavior when interacting with the pedestrian.** Robots and pedestrians with the same color indicate the same time. The black dotted lines indicate the time of closest proximity.

We run our robot five laps each in three different environments, with a control policy fine-tuned by our methods and two selected baselines. For more details regarding the experimental setups, please refer to the appendix C.1. Table 1 shows quantitative analysis with our method and selected the three strongest baselines (The measured metrics are shown on the caption).

Our method has the best scores for all metrics. In particular, our method reduces IDV and NCO by more than 50% against the strongest baseline and increases SPL and STL by about 10 %. CP and CO are improved by adding a negative reward for collisions detected by the mechanical bumper sensor. Although we penalize the collision in the model-based objective, it is based on the estimated poses, which is less accurate than using the bumper sensors. Hence, our learned Q values significantly improve on CP and CO. The reward design for the pedestrians can help prevent the robot from stacking in front of the pedestrians, which may worsen STL. Moreover, online learning could improve basic goal-reaching performance and reduce human interventions. These positive aspects suppress the undesired deviations and significantly improve the SPL.

To evaluate how well our method behaves around humans, we also conduct additional experiments with twelve human subjects. We recruited twelve subjects from among graduate students, visiting scholars, and staff members on campus. We consider the balance to be as diverse in gender (6 male, 6 female), professional-level (7 student, 5 non-student), and origin (4 North America, 5 Asia, 2 Europe, 1 Latin America) as possible. For more details, please refer to the appendix C.2. Following [72], subjects were asked nine qualitative questionnaires [72] with a 7-point Likert scale from "Not at all" to "Very much" after each experiment. To mitigate bias in favor of the first method, we present the questionnaires [72] in Fig. 3 before running evaluations with the first method.

Figure 3 shows the means and standard deviations of the scores from this study. Note that we flip the score for the last question such that higher is better for the entire bar plot. **Residual RL** often violates the intimate distance and occasionally collides with the human subject. As the result, **Residual RL** has the worst score. **SACSoN with fine-tuning** shows similar behavior in navigation and the closest score to our method. However, our method performs better overall. The statistical significance of our method is confirmed in four questions, (4), (6), (8) and (9) on the t-test with $p < 0.05$.

Figure 4 visualizes the robot behavior when the pedestrian and robot pass each other. In these time lapse illustrations, the color of the pedestrian and robot indicates the timestep (i.e., a yellow pedestrian and a yellow robot indicate the same point in time). In addition, we use a black dotted line for the robot and the pedestrians when robots and people are in closest proximity. In all cases, the pedestrian gets stuck in front of or behind the robot running the baseline method (right), and the robot penetrates the intimate distance for an extended length of time. Especially in case A and D, the baseline fails to reach the goal position and collides with obstacles. With our method, an

evasive maneuver is initiated at an early stage and succeeds in smoothly passing a pedestrian without getting stuck (left). Although the robot is close to the pedestrians when passing, our control policy minimizes how long the robot penetrates the pedestrian's intimate distance.

Figure 5 shows the time lapsed images when avoiding the small unseen objects and the uneven rubber mat. Our method naturally avoids colliding with the small objects, though this presents a challenge for the initial SACSoN policy. In addition, our methods avoids traveling on the uneven mat since we give a negative reward $C_s$ during online training. Please see our supplemental materials for more details.

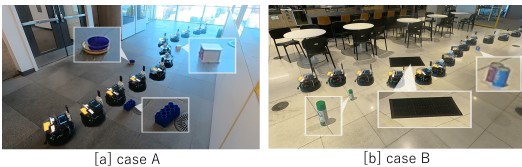

[a] case A  [b] case B

Figure 5: **Visualization of the robot behavior of avoidance for small obstacles and uneven floor mat.**

**Interventions during online training:** Reducing human interventions is important in online training to enable autonomous adaptation in the real world. However, it is known that the data distribution shift between offline and online training causes performance degradation, leading navigation failures and a lot of interventions to keep online learning in navigation. To evaluate the online training process, we count the number of human interventions in each navigation loop for **Q2**.

Figure 6 shows mean and standard deviation of the interventions in three environments. Our method gradually decreases the number of interventions during online training, and it almost reaches to zero at 15 laps. It means that SELFI can consistently improve the performance without degradation. On the other hand, the number of interventions increase for the baselines over the first few laps. Afterwards, the baselines decrease the number of interventions. However, the baselines still need a few human interventions to complete navigation.

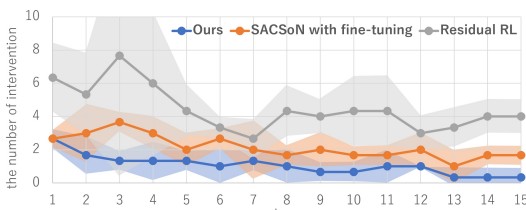

Figure 6: **The number of intervention on online learning in three different environments.** The lines indicate the mean and the areas indicate the standard deviation. The horizontal axis indicates the number of times the robot laps the loop reference.

Please see additional evaluation with more baselines in Appendix E and supplemental videos.

## 6 Discussion

We proposed an online self-improving method, SELFI, to quickly fine-tune a control policy pre-trained with model-based learning. SELFI combines model-based learning and model-free RL its training objectives to take advantage of the best parts of both approaches. The same objectives used in offline learning are introduced into online learning to stabilize the learning process. The performance of the pre-trained policies are improved via Q-functions from online model-free RL.

In the evaluation, SELFI was implemented to fine-tune the SACSoN policy [5] for vision-based navigation. SELFI enables us to quickly learn complex robotic behavior, such as pre-emptive collision avoidance for pedestrians, collision avoidance for the small and transparent obstacles, and preferences for traversing on smooth surfaces. These behaviors are difficult to learn on offline training due to the modeling errors and data distribution shift. In addition, compared to various baseline methods, SELFI did not require much human intervention during online learning. The performance of the trained control policy by SELFI is also visualized in the supplemental materials.

While our method enables us to quickly fine-tune a pre-trained policy, it has some limitations. For effective online learning, the balance between objectives from model-based learning and the learned Q-function is important, but this balance cannot be predicted in advance and requires some trial-and-error with real robots. Although the reward for socialness is given only for intimate distance violations, human-in-the-loop online learning with human evaluations can lead to better behavior. And a more diverse multiple public datasets on offline training is practically required for robustness.

## Acknowledgments

We consulted the Committee for Protection of Human Subjects at our home institute, and it was determined that the study does not meet the definition of research with human subjects as outlined in Federal Regulations at 45 CFR 46.102. This research was supported by Berkeley DeepDrive at the University of California, Berkeley, and Toyota Motor North America. And, this work was partially supported by ARL DCIST CRA W911NF-17-2-0181 and ARO W911NF-21-1-0097. We thank Qiyang Li and Mitsuhiko Nakamoto for advising the mathematical formulation. We would like to express their gratitude to Roxana Infante, Ami Katagiri, Pranav Atreya, Stephan Allenspach, Lydia Ignatova, Toru Lin, Charles Xu, Kosuke Tahara, Katsuhiro Kutsuki and Catherine Glossop for their valuable assistance in evaluating SELFI.

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

# Appendix

## A  Robotic system

For online learning in the real-world, we build a vision-based navigation system that uses a topological graph of the environment, where nodes denote visual observations and edges denote connectivity.

**Hardware setup:** Figure 7[a] shows the overview of our prototype robot. We use an omnidirectional camera to observe $\{I_t\}_{t=-M\ldots0}$ and $I^g$. This allows us to observe a $360°$ view for capturing the pedestrians even behind the robot. The robot is equipped with an NVIDIA Jetson Orin AGX onboard computer, which runs inference of trained models at 3 Hz. We use two additional cameras to estimate visual odometry and to detect long-term localization fiducials, following the setup of Hirose et al. [5]. We use an IMU to measure *bumpiness* and uneven terrain and a bumper sensor to detect collisions. In addition to on-robot compute, we use a workstation for fast, online training. The workstation is equipped with an Intel i9 CPU, 96GB RAM, and an NVIDIA RTX 3090ti GPU.

Figure 7[b] shows small obstacles, which we place for online learing(left) as well as our evaluation(right). In our evaluation, we place the obstacles, which is not seen in online training.

**Navigation system:** Similar to [21, 61, 73], we construct our vision-based navigation system using a topological memory. We update the goal image $I^g$ based on localization in the topological map to navigate towards a distant goal position. Before deployment, we collect the map by human teleoperation and record the subgoal images and the corresponding global goal poses at 0.5 Hz as $\{I_i^g, p_i^g\}_{i=0\ldots L}$ along the robot's trajectories. Here, $L$ indicate the number of the nodes in the topological map. During inference, we estimate the global robot pose $p$ and decide the closest node number $i_c$ as the current node by $i_c = \arg\min_i \|p_i^g - p\|$ and feed $I_{i_c+1}^g$ as the goal image $I^g$. We estimate $p$ with incorporating visual odometry and AR markers as shown in the appedix.

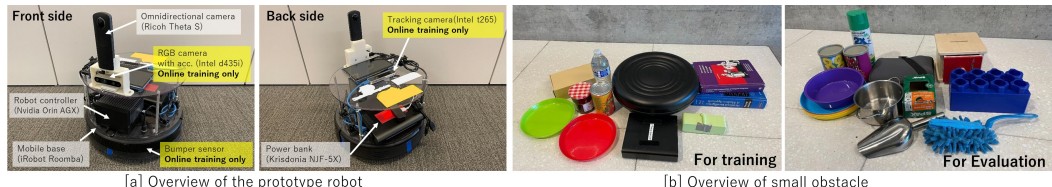

[a] Overview of the prototype robot          [b] Overview of small obstacle

Figure 7: **Overview of the prototype robot(left) [74] and small obstacles(right).**

## B  Evaluation setup

We evaluate following baselines in addition to our proposed method for comparative evaluation.

**Sampling-based motion planning:** This baseline generates fifteen motion primitives [75, 76] at every time step and selects the best one considering goal reaching, static and dynamic obstacles such as the pedestrians. To control the robot, we give the velocity commands corresponding to the selected motion primitive. The details are shown in the Appendix G.

**TD3+BC→TD3 [18, 38]:** This baseline uses TD3+BC [38], an offline RL method, to train the encoder, actor, and critic offline. During online training, we fine-tune the pre-trained actor and critic with TD3 [18] while freezing the encoder.

**FastRLAP [13]:** FastRLAP employs the pre-trained encoder from offline RL and trains the critic and the actor from scratch online while freezing the encoder. Different from the original FastRLAP, we use TD3+BC [38] and TD3 as offline and online learning algorithms, respectively.

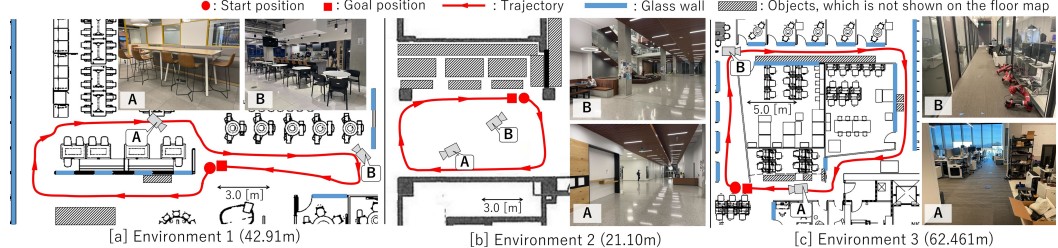

[a] Environment 1 (42.91m)      [b] Environment 2 (21.10m)      [c] Environment 3 (62.461m)

Figure 8: **Three environments on online training and evaluation.** We conduct online training in three different challenging environments, [a] the open space facing restrooms, elevator hall and café space, [b] the entrance hall with many pedestrians, and, [c] the office area with narrow corridors. [a] and [c] have many glass walls, which are difficult for collision avoidance and cause lighting condition changes.

**Residual RL [14]:** In residual RL, the policy is given by the sum of a base policy and a learned policy, $a = \pi^{\text{base}}(s) + \pi^{\text{RL}}(s)$, where $\pi^{\text{base}}$ is the pre-trained control policy and $\pi^{\text{RL}}$ is the actor trained with online RL (TD3). We evaluate two choices for $\pi^{\text{base}}$: (1) the pre-trained control policy maximizing only $\sum_{t=0}^{H-1} \hat{r}_t^{\text{pose}}$ to simply move towards the goal position, and (2) the pre-trained SACSoN policy, maximizing the total objective $J_{\text{sacson}}$. We label the latter as **Residual RL**[†]. Residual RL provides an alternative way to combine prior policies with online model-free RL, and therefore represents a natural prior method for comparing with SELFI.

**SACSoN with fine-tuning [5]:** This baseline trains the entire control policy by maximizing the SACSoN objective $J_{\text{sacson}}$ on the HuRoN dataset, and then fine-tunes the actor online by maximizing $J_{\text{sacson}}$ again. The online objective does not use the additional (non-differentiable) online reward terms $r$.

All learning-based baselines use the same network structure, except that single-step methods (all except **Ours** and **SACSoN with fine-tuning**) predict only a single action $a$ rather than a sequence $\tau$. Unless specified, all RL-based methods use the same reward.

We conduct our experiments in three challenging environments in Fig. 8, which are in different regions of the same building. Environment 1 is an entrance and café area, which naturally has a lot of pedestrians. The environment's lighting conditions and furniture placement change significantly over time. This environment also contains glass walls and chairs with thin legs, which can be difficult to detect as obstacles. Environment 2 is the entranceway to an office building, also with significant pedestrian traffic. Environment 3 is a loop through several hallways, desk areas, and working spaces. Pedestrians are less frequent in this environment, but the corridors are narrow and require avoiding difficult static obstacles such as glass walls, and present a challenge in avoiding pedestrians in confined spaces.

In these three environments, we design the looped trajectories, which the last node is same pose as the initial node, as shown by red lines in Fig. 8, and feed the first goal image when arriving at the last node to continuously train the control policy online. We conduct online learning while rotating these loops and stop training when the robot run 15 laps or two hours. During online training, we randomly place small objects shown in Fig. 7[b](left) to learn the collision avoidance behavior for the small objects. Since Environment 1 includes many challenges for our task, we evaluate all methods in this environment and conduct the overall evaluations with the selected baselines in three environments.

## C    Evaluation with human participants

For **Q1**, we conduct two types of experiments, 1) comparative evaluations with relevant baselines and 2) human subject experiments to evaluate how well our methods behave. For 1), we conduct the experiments in the "organized" environments for reproducible comparisons. For 2), we conduct

our experiments in the "unorganized" environments to obtain various evaluations of each human subject's senses. We explain the details of the experimental setups in each experiment.

## C.1 Reproducible comparisons

It is difficult to conduct extensive and reproducible comparisons in highly populated natural environments because pedestrians are walking in and out of the scene at random intervals. Therefore, to conduct a more controlled and reproducible comparison with each of the baselines, we focus on the "organized" environments.

Before the experiments, we instruct the human participants to follow pre-defined trajectories (as repeatably as possible) to have consistent testing conditions. However, if the robot interferes with a person's path, the pedestrians were asked to slow down or stop as needed. If the robot can not give way to the pedestrians, the pedestrians change their path as they deem fit, to interact with the robot and then come back to the original trajectories. All subjects are asked to follow these instructions across the experiments. We conduct the experiments across different days, as well as times of day, mimicking the range of lighting conditions and changes in environment layout that a robot would experience over several days. In each experiment, we randomly place the small unseen objects shown in Appendix A into the scene to increase the clutter in the scene and evaluate the collision avoidance performance. However, we conduct the experiments under approximately equivalent conditions for each method.

## C.2 Human subject experiments

To incorporate various human subjects into our evaluation, we ask the human subjects to interact with the robot and evaluate its behaviors without specifying specific ways for the subjects to interact with the robot. This prevents biased behavior and insights from the human subjects. Additionally, we ask the human subjects to have similar interactions with the robot across all methods to maintain fairness in the evaluation. To assure the behaviors of the pedestrians were as natural as possible, we did not always observe and follow the robot during online training. Subjects were not told which method was being used for each trial. Before the evaluation, we explain the rough robot route from the start to the goal position.

## D  Weighting objectives on online training

Similar to the other learning algorithms, our approach needs to find the best balance of each objective on online training. In our case, we first perform online learning with small weights for the learned objective $\bar{Q}(s, a)$ and gradually increase the weights. The small weighting for the learned objective makes online learning more stable and facilitates the analysis of learning results. Once we find a good weight in one environment, we use the same value in the different environments.

## E  Pre-training with data from the target environment

When robots operate continuously in the same environment, they can directly collect and utilize a large amount of in-domain data. In the supplemental evaluation, we show that a large offline dataset of interactions from the target environment can boost the performance of our method during online training. Specifically, we use offline RL to train our actor and critic on the collected dataset from the target environment on top of SACSoN. Then, we fine-tune both networks online by SELFI.

During offline RL training, we leverage the collected dataset from our study. While finding good hyperparameter for online learning, we collected a dataset of 50 hours of experience in Environment 1. Figure 9 shows the number of intervention for each loop in Environment 1. Even at beginning, our method leverages a large dataset, **Ours + prior data** only needs one intervention and can navigate the robot without any interventions after 5 laps, because of faster training with the pre-trained actor and critic on the large dataset. Similar to Table 1, Table 2 shows the mean value of the selected

metrics on 5 laps in the Environment 1. **Ours + prior data** shows a remarkable gap against **Ours** in every metric except SPL. Since **Ours + prior data** takes a larger deviation from the original path to avoid violating the intimate distance of close pedestrians (to decrease IDV), SPL is slightly worsen.

In addition to **Ours + prior data**, we show the results of the all other baselines including **FastRLAP**, **TD3+BC→TD3** and **Residual RL**[†] in Fig. 9 and Table 2. **FastRLAP** and **TD3+BC→TD3** can also improve the performance of the control policy during online training. However, online training for two hours or 15 laps is not sufficient, and these methods require many interventions. Surprisingly, applying **Residual RL**[†] with the pre-trained SACSoN policy as $\pi^{\text{base}}$ actually *decreases* in performance during online training, giving worse performance than **Residual RL**. We hypothesize that in this case $\pi^{\text{RL}}$ must learn a *copy* of the SACSoN policy to predict the result of a particular action, saturating the capacity of the network due to the base policy's high complexity. We find that **Residual RL** and **SACSoN with fine-tuning** are the strongest baselines. And we think that Sampling-based motion planning can be the proper baseline from non learning-based approach. Hence we prioritize these three baselines in our further evaluations in Sec. 5.

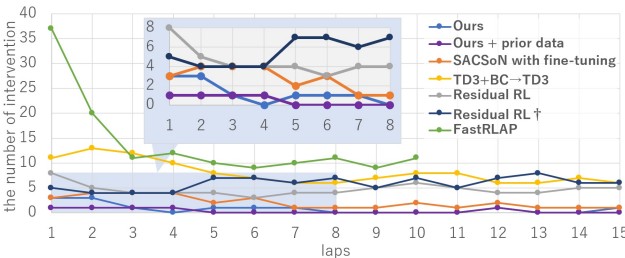

Figure 9: **The number of intervention on online learning in Environment 1.** Ours + prior data indicates our method leveraging large dataset in same environment. Residual RL[†] uses the SACSoN policy as $\pi_{\text{base}}$. The horizontal axis indicates the number of times the robot laps the loop.

Table 2: **Closed-loop Evaluation of trained control policies at Environment 1.** Ours + prior data indicates our method leveraging large dataset in same environment. Residual RL[†] uses the SACSoN policy as $\pi_{\text{base}}$. ∗ indicates the use of the ground truth goal pose to generate velocity commands.

| Method | IDV [s] ↓ | CP ↓ [#] | Int ↓ [#] | SPL ↑ | STL ↑ |
|---|---|---|---|---|---|
| Sampling-based motion planning∗ | 23.909 | 1.800 | 0.400 | 0.839 | 0.678 |
| TD3 + BC→TD3 | 34.632 | 0.800 | 6.600 | 0.773 | 0.383 |
| FastRLAP | 43.623 | 2.000 | 10.000 | 0.702 | 0.306 |
| Residual RL | 21.754 | 4.200 | 3.000 | 0.780 | 0.568 |
| Residual RL[†] | 39.627 | 2.800 | 9.400 | 0.598 | 0.466 |
| SACSoN with fine-tuning | 19.534 | 0.200 | 1.000 | 0.788 | 0.549 |
| Ours | 10.712 | **0** | 0.600 | **0.932** | 0.699 |
| Ours + prior data | **6.610** | **0** | **0.200** | 0.900 | **0.713** |

# F   Robot pose estimation with AR markers

For reward calculation, we first estimate the robot's global position $p$. Using $p$, we can get an estimate of $g$, the goal position in the robot's local frame, which we can directly use for our reward calculation. Additionally, $p$ is useful to localizing the robot's position in a topological map, which provides our policy the current subgoal image $i_c$.

For localization, we mount a tracking camera, Intel T265 on our robot that measures visual odometry. However, the Intel T265 can not maintain the same globalframe before and after rebooting due to battery replacement. In addition, the shaky robot motion and insufficient visual features deteriorate the accuracy of visual odometry. To have same global frame and to get a more accurate estimate

of the robot's global position, we place one AR marker every 15 [m] along the robot's trajectory and suppress the localization error for stable online learning.

Figure 10 shows an overview of our localization system using AR markers. Before starting online learning, we collect $\{T_i^g, T_i^{AR_g}\}_{i=1...O}$ in conjunction with a topological map. $T_i^g$ is the robot position matrix computed from visual odometry and $T_i^{AR_g}$ is the position matrix of $i$-th AR marker in the robot's local frame when detecting $i$-th AR marker. Here, we assume the robot's motion during teleoperation is smooth enough for us to have an accurate estimate of $T_i^g$ from visual odometry. $O$ is the number of AR markers on the robot's trajectory. When detecting the $i$-th AR marker during online learning, we calculate the correlation matrix $T_i^c$ to estimate $T$ from the visual odometry $T_t^{vo}$:

$$T = T_i^c \cdot T_t^{vo}. \tag{4}$$

When detecting the $i$-th AR marker during online learning, $T$ can be defined as follows,

$$T = T_i^g \cdot T_i^{AR_g} \cdot (T_i^{AR_r})^{-1}. \tag{5}$$

Here $T$ is on the same global coordinate as $T_i^g$. Hence, we can obtain $T_i^c$ by calculating $T_i^g \cdot T_i^{AR_g} \cdot (T_t^{vo} \cdot T_i^{AR_r})^{-1}$. During online learning, we update $T_i^c$ every time we detect an AR marker, and we use $T_i^c$ to calculate $T$ until the robot detects the next AR marker. The robot pose $p$ is uniquely calculated from the robot pose matrix $T$. And, the goal pose $\hat{g}$ on the robot local coordinate can be uniquely calculated from $T^{-1}T_g$, where $T_g$ is the global goal pose matrix for the goal image $I^g$.

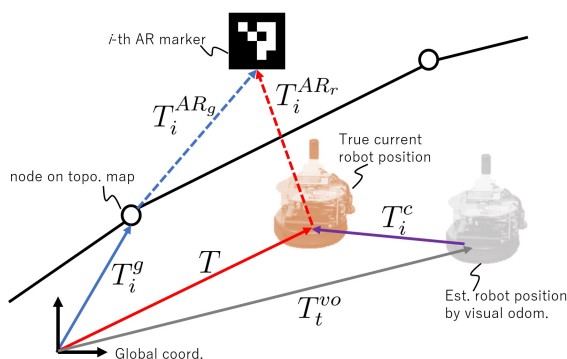

Figure 10: **Overview of robot's position estimation with AR markers.** We update the correlation matrix $T_i^c$ when the robot detects an AR marker, and we apply it to suppress the error from the noisy visual odometry $T_t^{vo}$ until the robot detects the next AR marker.

While we use AR markers due to our design decision using only an RGB camera and limitations of the Intel T265, it is important to note that the usage of AR markers is not mandatory. Our system setup serves as just one example. If potential users relax the restriction of using only the camera, they can leverage other sensors such as GPS, LiDAR, and/or depth cameras to estimate the robot's pose without requiring AR markers. It's worth mentioning that we can run our system without AR markers during inference using other vision-based localization techniques [21, 25, 61, 74, 77], but we use AR markers to identify the corresponding goal image ID, focusing on evaluating the trained control policy.

## G   Details of sampling-based motion planning

We implemented sampling-based motion planning as the baseline to bridge the learning-based approach with broader robotic motion planning. We generated 15 motion primitives assuming steady linear and angular velocity commands for 8 steps (2.664 s), which is the same horizon as our method and the strongest baseline, SACSoN. The pairs of linear and angular velocity commands are $(v_s, \omega_s) = (0.0, 0.0), (0.2, 0.0), (0.2, 0.3), (0.2, 0.6), (0.2, 0.9), (0.2, -0.3), (0.2, -0.6), (0.2, -0.9), (0.5, 0.0), (0.5, 0.3), (0.5, 0.6), (0.5, 0.9), (0.5, -0.3), (0.5, -0.6), (0.5, -0.9)$. We selected these 15 motion primitives by balancing computational load and navigation performance.

By integrating these velocity commands for 8 steps, we obtained 15 trajectories such as $\{\{{}^s p_i^j\}_{i=1...8}\}_{j=1...15}$, where ${}^s p_i^j$ is the $i$-the virtual robot pose on the $j$-th motion primitive. To select the best motion primitive, we calculated the following cost value for each primitive.

$$J_s^j = \min_i (p_{i_c+1}^g - {}^s p_i^j)^2 + C_{ob} + C_{ped} \tag{6}$$

Here, $p^g_{i_c+1}$ indicates the next subgoal pose. The first term on the right-hand side calculates the squared errors between all 8 poses in the $j$-th motion primitive and the goal pose and selects the minimum one to evaluate the goal-reaching performance. $C_{ob}$ is a constant value used to filter out trajectories that collide with static obstacles. We calculate $C_{ob}$ as follows:

$$C_{ob} = \begin{cases} 1000.0 & (\text{if } d_s < r_r) \\ 0.0 & (\text{otherwise}) \end{cases}, \tag{7}$$

where $d_s$ is the minimum distance between all 8 poses in the $j$-th motion primitive and the estimated point clouds corresponding to the static obstacles. Similar to SACSoN [5] and ExAug [25], a collision is determined when the distance between the static obstacle and the robot is less than the robot's radius $r_r$. To ensure fair evaluation with other methods in vision-based navigation that only uses an RGB camera, we utilize estimated point clouds from the current observation of the RGB camera. Additionally, $C_{ped}$ is a constant value used to filter out trajectories that violate the intimate distance with pedestrians. Following SACSoN [5], we predict the future trajectory of pedestrians as $\{p^{ped}_i\}_{i=1...8}$ and assess whether each motion primitive violates the intimate distance or not.

$$C_{ped} = \begin{cases} 1000.0 & (\text{if } d_{ped} < 0.5 + r_r) \\ 0.0 & (\text{otherwise}) \end{cases}. \tag{8}$$

Here, $d_{ped} = \min_i \text{dist}(p^{ped}_i, {}^s p^j_i)$, and dist() is the function used to calculate the distance between two poses. The values of $C_{ob}$ and $C_{ped}$ are set to 1000.0, which is much larger than the first term on the right-hand side in Eqn 6, to filter out inappropriate motion primitives during the selection process. We choose the motion primitive with the minimum $J^j_s$ and assign the corresponding velocity commands $v_s$ and $\omega_s$ to control the robot during navigation. Note that we utilize the same point cloud estimator as well as the pedestrians' trajectory predictor as our method and SACSoN to have fair comparison.

Figure 11 shows the implemented sampling-based motion planning. Since pedestrians are not present in [b], this baseline selects the motion primitive that moves toward the goal position. However, in [a], where the pedestrian's predicted trajectory crosses between the robot's current position the goal position, the baseline selects a motion primitive that maintains a sufficient distance from the pedestrian's predicted trajectory. After the pedestrian passes through the area, the baseline will select a trajectory towards the goal. In Fig. 11, the blue lines with the "x" markers are the motion primitives and the black dots are the estimated point clouds.

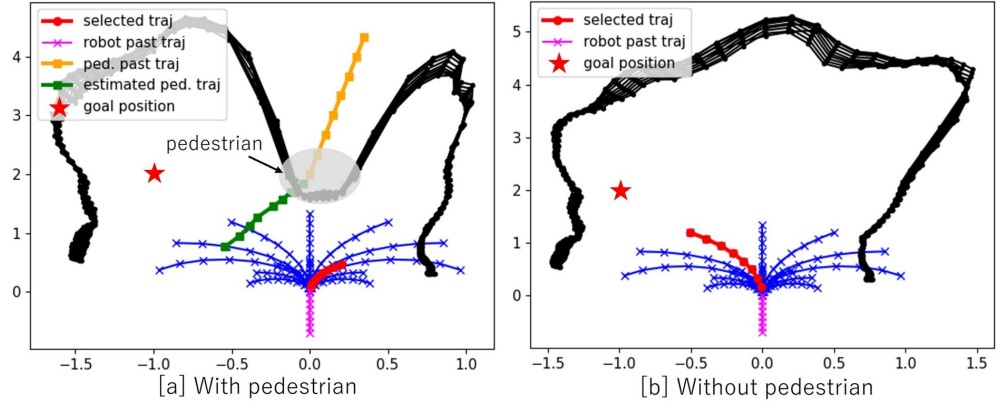

Figure 11: **Overview of sampling-based motion planning.**

## H  Details of network structure

Each raw image from the robot's omnidirectional camera is a front- and back-side fisheye image stiched side-by-side. We resize the stiched images and concatenate them in the channel direction

such that the resulting image data $6 \times 128 \times 128$. To extract temporally consistent features in the observation and goal image, we channel-wise concatenate the goal image and a history of observation images as $48 \times 128 \times 128$ input and feed it into the encoder, $g_\phi$. Note that 6 channels are for current observation, $6 \times 6$ channels are for the history of past observations, and the last 6 channels are for the goal image. The encoder has eight convolutional layers with batch normalization and ReLU activation function in each layer to extract a 512-dimensional feature vector. The feature vector is then fed into the actor $\pi_\theta$, a full-connected MLP.

The extracted features for the actor are concatenated with the previous action commands $\tilde{a}$ to handle the deadzone, which is a result of system delay and backlash in the robot hardware. The actor $\pi_\theta$ has three fully connected layers with batch normalization and ReLUs to generate $\tau$. The last layer has a hyperbolic tangent function to limit the velocity commands within upper and lower boundaries.

We concatenate $\tau$ and the extracted features from the image encoder and feed it into the critic to estimate $\bar{Q}$. The critic is designed with five fully connected layers, batch normalization, and ReLUs. The last layer has a linear function instead of ReLUs to allow us to estimate negative values.

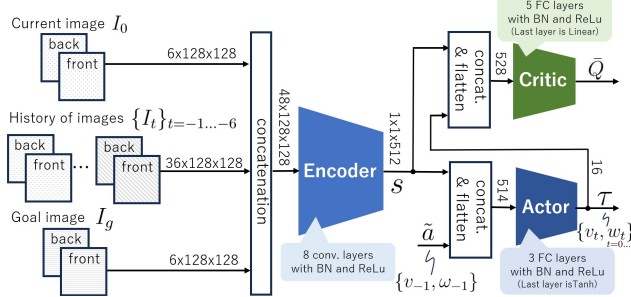

Figure 12: **Overview of network structure in our implementation.**

