# OpenReview forum: "SELFI: Autonomous Self-Improvement with RL for Vision-Based Navigation around People"
_robot-learning.org/CoRL/2024/Conference — CoRL 2024_

### Official Review · Reviewer_1Aui · 2024-06-29
**Good paper, but needs clarification**

**Originality:** 4
**Technical Quality:** 4
**Clarity Of Presentation:** 2
**Potential Impact:** 3
**Recommendation:** 3
**Confidence:** 5

**Review:**

**Post rebuttal edit:**
The authors did not upload the revised paper or supplementary material. In addition, their response to my review was vague and did not address most of my concerns. Given the presentation flaws in the introduction and other sections, and the fact that the quality of this paper is indeed solid, I changed my recommendation to "weak accept". I don't mean to prevent this paper from being accepted if this is what the AC and other reviewers recommend, I just want to emphasize the importance of addressing the reviewers' comments in rebuttal, since reviewers spent time and efforts reading your paper and writing the review.
**End of post rebuttal edit**

Overall, I believe the paper is a good contribution to vision-based social navigation, as well as the CoRL community. The main advantages of this paper are detailed below.
- This paper tackles a difficult and important robotics problem. Vision-based navigation significantly reduces the cost of sensors and errors from perception modules, while navigation among people in a socially aware manner is a key step for robots to operate in everyday environments.
- The methodology that combines model-based method and RL makes sense to me. In social navigation, model-based methods can solve part of the problem, and it is reasonable to use RL to learn the rest of the part.
- The experiments were conducted in real environments, and a user study is conducted with real users. This is important to show the potential of the proposed work in deployment in the wild.

However, I find the clarity and accessibility of presentation, especially in introduction and experiment sections, could be improved. Please proceed to “Questions for rebuttal” below.

**Quality Of The Limitations Section:**

3

**Questions For Rebuttal:**

- For readers who do not know SACSoN, the introduction is hard to digest. Please consider filling in the relevant background so that this paper can better stand on its own. In addition, some other parts are unclear:
  - Citation is needed in line 22 after “Model-based control… in the scene.”
  - In line 42-43, I don’t understand why RL can work as expected “as long as…small values.”
  - In line 68, please point out what type of model-based learning is used. “Model-based learning” is a very general phrase.
- The authors defined the reward $r$ in line 214. But $\hat{r}$ is mentioned but not clearly defined in line 194. Also, what is $\overline{r}$ used in the experiments?
- In online learning stage, are pedestrians real or arranged? If real, how to deal with safety issues when the robot fails?
- In experiments, what is the range of robot’s travel distance, the range of the number of pedestrians, and number of environments/scenarios for training and testing?
- In line 279, why “our method performs better overall”?
- Are human intervention signals used for training? If not, how could this be done in future work?

Minor problems:
- In Sec. 2,
  - the authors may consider citing the following papers as examples of model-based methods:
    - Van Den Berg et al, "Reciprocal n-body collision avoidance," Robotics Research, 2011.
    - Mavrogiannis et al. "Winding through: Crowd navigation via topological invariance," RA-L 2022.
  - And consider citing the following papers as RL methods:
    - Liu et al, "Decentralized Structural-RNN for Robot Crowd Navigation with Deep Reinforcement Learning," ICRA 2021.
    - Xie et al, "DRL-VO: Learning to Navigate Through Crowded Dynamic Scenes Using Velocity Obstacles", T-RO 2023.
- Table 1 should be moved to page 7.
- In Fig. 4, the yellow robot is hard to see in the left 2 subfigures, and the green human is missing in the right two subfigures.

**Robotics Focus:**

4

**Summary Of Paper:**

This paper proposes a model-based + online RL pipeline for vision-based social navigation. The proposed method overcomes the inaccurate models of model-based method and data inefficiency of end-to-end RL. The experiments evaluated both navigation metrics and subjective feedback from a user study. The results show that the proposed method outperforms baselines in terms of navigation metrics, user feedback, and number of human interventions during online learning.

**Summary Of Recommendation:**

This paper investigates an interesting topic with novel methodology and solid experiments. It would be good if the authors could improve the clarity of presentation.

---

### Official Review · Reviewer_5y57 · 2024-07-05
**Interesting and good approach**

**Originality:** 3
**Technical Quality:** 4
**Clarity Of Presentation:** 4
**Potential Impact:** 3
**Recommendation:** 3
**Confidence:** 4

**Review:**

The paper is generally well-written, with a proposal that is both intuitive and well-grounded. Its application to social navigation effectively demonstrates the capabilities of online learning, showing how real-time adaptation can enhance autonomous systems' performance in dynamic social environments. The proposed approach of combining offline and online policies through the objective function is innovative and clearly presented.

Strengths:
1. The definition of the Q function $Q(s, \tau) = J(s, \tau) + \bar{Q}(s, a)$ is clear. The combination is also sensible, and its benefits of stabilizing the online learning process are evident.

2. The proposed framework SELFI is described clearly, and the choices made are well-documented. For instance, the use of the Huron dataset alongside the online dataset for the training process effectively mitigates the risk of overfitting on online data. Additionally, the design of the reward structure for the online process is meticulously detailed, enhancing the framework's robustness and applicability.

3. The results obtained appear to be strong in comparison to existing approaches.

Weakness:
1. The evaluation section (Section 5) is largely descriptive and lacks in-depth analysis or explanations of the obtained results presented in Table 1, Figure 3, Figure 5 and Figure 6. The authors primarily reiterate the presented data without providing any detailed analysis or insightful interpretation of the results. For example, "Table 1 shows quantitative analysis with our method and selected the three strongest baselines. Our method has the best scores for all metrics.", "Figure 3 shows the means and standard deviations of the scores from this study", "Figure 5 shows the time lapsed images when avoiding the small unseen objects and uneven rubber mat. Our method naturally avoids colliding with the small objects, though this presents a challenge for the initial SACSoN policy.", "Figure 6 shows mean and standard deviation of the interventions in three environments. Our method gradually decreases the number of interventions during online training, and it almost reaches to zero at 15 laps." The authors could provide additional insights to interpret the results and further strengthen the case for SELFI.

2. Limitations can be improved.

**Quality Of The Limitations Section:**

2

**Questions For Rebuttal:**

1. I would prefer if the authors could improve their discussion and evaluation of the results. Currently, much of the paragraph in the evaluation section is a detailed description of the evaluation by human rating. If paper length is a consideration, these details could be relegated to the appendix since much of it is similar to existing works. A more detailed analysis of the results and their interpretation would significantly strengthen this work. (e.g., Why CP, CO, SPL, STL are significantly better than the other baselines?)

2. The implementation of SELFI uses a frozen encoder during the online phase, which implies that the online inputs need to be relatively similar to those seen in the offline dataset. Otherwise, the extracted features may not be accurate. Are there any precautions taken to address this? If the deployed environment differs from the training environment, or if there are new elements (e.g., new types of obstacles) that were not encountered during training, does the approach still perform well?

3. SELFI uses a model-based approach to compute the SACSoN objective even during the online phase. While this objective is beneficial for the initial training phases, is it necessary to maintain it throughout the entire online deployment? As with all model-based approaches, the performance of the control policy depends on the accuracy of the model, and it might be beneficial to reduce such dependencies over time. Have the authors considered the use of, say, a decay on the offline objective to gradually phase out the dependency on the model and transition to a completely model-free approach? Is this approach feasible? Or rather, is there a reason that the offline objective needs to be maintained throughout?

**Robotics Focus:**

4

**Summary Of Paper:**

The paper introduces an online learning method SELFI that leverages online experiences to rapidly fine-tune pre-trained control policies. While closely related to residual RL, SELFI considers the linear combination of the offline and online policies through the objectives (i.e., Q-values) rather than their direct actions. The incorporation of the offline model-based learning objective into the online learning process stabilizes the training process and allows quick adaptation to the operational environment.

**Summary Of Recommendation:**

The paper is well-written and the approach, though simple, is well-supported theoretically and empirically.

---

### Official Review · Reviewer_agf9 · 2024-07-11
**SELFI Review**

**Originality:** 3
**Technical Quality:** 4
**Clarity Of Presentation:** 5
**Potential Impact:** 3
**Recommendation:** 4
**Confidence:** 5

**Review:**

Summary Of Contribution: The paper focuses on mobile ground robots that move in environments with humans around them. Here especially, the focus on safe AND social behavior is given. The authors provide a Hybrid Learning Approach, their method SELFI effectively combines offline model-based learning with online model-free reinforcement learning, leveraging the strengths of both paradigms.

Summary Of Strengths:
- The paper addresses a critical challenge in the field of social robotics, particularly when operating in close proximity with humans
- The Hybrid Learning Approach effectively combines offline model-based learning with online model-free reinforcement learning, leveraging the strengths of both paradigms.
- The authors present possibilities on how to allow for rapid fine-tuning of pre-trained control policies in real-world environments, improving performance in a relatively short amount of time.
- The SELFI method demonstrates good behavior in collision avoidance. The author are able to demonstrate that especially with small and transparent objects, which are often challenging for robots.
- Ultimately, the framework presented in this paper enables socially compliant behaviors around pedestrians, such as pre-emptive avoidance and maintaining appropriate distances.

Summary Of Limitations:
- It looks like that the authors experimented a lot in finding the right balance between model-based and model-free objectives during online learning. RIght now, for the reader  or somebody who wants to reimplement it it can be challenging and may require trial-and-error. Are there any suggesting on how to overcome this?
- A big limitation is the dependence on AR Markers for localization, which may not be practical in all environments. What would happen if the Localization is not that good or even noisy?

**Quality Of The Limitations Section:**

1

**Questions For Rebuttal:**

Feedback For Improvement Or Clarification:
- I think that it would be helpful to provide more guidelines or automated methods for balancing the model-based and model-free objectives during online learning to reduce trial-and-error. Maybe it is possible to enter some more information for the authors on how to do that or provide even pre-trained models?
- Currently, I am missing  a clarification on the reward design process and how it adapts to different tasks and environments
- Furthermore, can you specify if your system can be adapted to different environments? Since you heavily rely on the AR markers, it is difficult to see how we move the system to a different environment
- The limitation section is currently missing. I would be good to have your own view on where the limitations of your paper are.

**Robotics Focus:**

4

**Summary Of Paper:**

The authors present a technique for vision-based navigation in mobile robot systems. They achieve this behavior by combining two types of RL based techniques. Firtly, the apply a model-based learning offline and then in a 2nd step additional RL based Online learning. This combination gives the authors to chance to apply it to a small-scale mobile robot and evaluate a navigation task around people. Further, the authors not only evaluate the quality of the navigation task but also evaluate the socialness by humans in order if they have the feeling that the robot is moving socially.

**Summary Of Recommendation:**

The paper is showing a nice method for visual based navigation especially in social situations. The paper is well written, adresses the methodology in a good way and shows real-world experiments - both qualitative and quantiative.

---

### Official Review · Reviewer_XMR4 · 2024-07-18
**Great paper with some confusion in the experiments**

**Originality:** 4
**Technical Quality:** 3
**Clarity Of Presentation:** 5
**Potential Impact:** 3
**Recommendation:** 4
**Confidence:** 5

**Review:**

Strengths

The paper is well presented with critical details included in the main text. The  proposed method is novel and impactful, and can further advance research on visual-based navigation. The offline-trained SACSON based on the HuRON dataset is already a model that performs visual navigation and pedestrian intent prediction quite well. It is exciting to see SELFI pushes this even further.

The authors also performed all experiments in real-world settings in both a controlled setting for quantitative analysis and a natural setting for qualitative analysis. So the evaluation is strong, particularly regarding handling small obstacles and uneven surfaces.

Weaknesses

It is unclear how the pedestrians behave in the controlled “organized” environments. Metrics such as IDV, CP imply that pedestrians are present in these experiments. Information related to the pedestrians’ behavior and density is missing. Additionally, the sampling-based baseline was not evaluated in the natural setting with the 12 participants. This raises questions on whether SELFI truly beats the sampling-based baseline in terms of social navigation.

Related to the first point, in the video, although the robot avoids pedestrians much better than Residual-RL and Sacson-finetuned. It still looks laggy and not so responsive. Interaction scenarios also only involve one or one group of pedestrians, so the proposed method might still be limited in very crowded scenarios.

It is mentioned in the limitation section. But in section 2 when comparing with Residual RL, the authors made a point about SELFI composing in the objective space rather than the action space. Although SELFI outperforms Residual RL, it is unclear whether this insight contributed to the superior performance, because similar confusion also seems to exist between the existing and learned objectives.

Minor:
“for for” on line 92


========================================================

Thank you for your rebuttal comments. I appreciate the addition of details regarding the experiment design with human participants. I also understand the difficulty of running experiments in the real world. With the authors' rebuttal, I don't think this paper has any noticeable weakness. I think I will bump my score up to strong accept.

**Quality Of The Limitations Section:**

3

**Questions For Rebuttal:**

What does the pedestrian setup look like in the “organized” environments?
Why is the sampling-based method not included in the natural evaluation setting with the 12 participants?

**Robotics Focus:**

4

**Summary Of Paper:**

This paper proposes a model-free online fine-tuning method on a model-based offline trained model to increase the robot’s capability in handling small obstacles, bumpy surfaces, and pedestrians.

**Summary Of Recommendation:**

This paper's quality is high, however it is confusing how humans are modeled in the “organized” environments setting, which raises questions on the quantitative analysis of the social navigation aspect. The sampling-based baseline is also missing in the qualitative analysis.

---

### Decision · Program_Chairs · 2024-09-04

**Decision:**

Accept

**Comment:**

Strengths:
- A critical and timely challenge is addressed.
- The proposed approach is technically sound and well-validated.
- The presentation is clear and the paper is well-written.

Weaknesses:
- There is some unclear parts in the evaluation setting.
- Dependency on a specific marker could be a limitation for exentions.